# UPER: Aligning Personalized Image Generation with Human Perception via Reinforcement Learning

## Abstract

Personalized image generation aims to synthesize novel scenes featuring a specific user-provided subject. However, state-of-the-art models often fail to preserve the fine-grained details that define a subject's unique identity, a critical flaw that limits their use in high-fidelity applications. This "consistency gap" arises from a misalignment between the model's learned similarity metric and nuanced human perception. To address this, we introduce **UPER** (**U**nifying **P**ost-training for **Per**sonalization), a post-training framework designed to align generative models with human preferences for detail consistency. UPER employs a two-stage process: it first refines the model's focus on the subject's core attributes via Supervised Fine-Tuning (SFT) on a dataset with cleaned background information. Subsequently, it optimizes the model using Reinforcement Learning (RL) with a novel composite reward function. The key component of this function is a new patch-based consistency metric that accurately measures subject fidelity using only pre-trained vision encoders, eliminating the need for expensive preference data collection. We apply UPER to the state-of-the-art OminiControl model. The results are unequivocal: in a blind user study with over 1,000 responses, images generated by our final model were preferred for their overall quality and subject consistency **89.3%** of the time over the strong baseline. Our work provides a robust and scalable solution to the detail-consistency challenge, paving the way for more faithful personalized generation.

## 1 Introduction

Large-scale diffusion models have achieved remarkable success in generating high-fidelity images from text descriptions (Rombach et al., 2022; Saharia et al., 2022; Esser et al., 2024). A pivotal frontier in this domain is personalized generation, where models are conditioned on a reference image to transfer specific subjects or styles into new creations (Ye et al., 2023; Tan et al., 2024). This multi-modal conditioning offers far greater precision than text alone, enabling high-value applications from virtual try-on (Han et al., 2023) to hyper-realistic product visualization for e-commerce and advertising (Zhang et al., 2024).

Despite this progress, a critical limitation persists: a failure to maintain detail consistency. As shown in Fig. 1, while models can replicate the general form of a subject, they often lose the specific textures, patterns, and structural nuances that define its unique identity. This "consistency gap" is particularly detrimental in commercial applications, where preserving brand logos, specific colorways, or unique material finishes is paramount. The issue is not merely a technical flaw but a fundamental alignment problem: the model's internal objective for "similarity" deviates from nuanced human perception. For instance, a model might prioritize matching the color of a shirt, while a human user cares more about preserving the logo printed on it. This misalignment stems from training paradigms that either lack diversity (self-generation) or rely on scarce, imperfectly paired data, causing the model to learn a generalized concept rather than specific details.

To bridge this perception gap, we turn to Reinforcement Learning from Human Feedback (RLHF), a powerful paradigm for instilling complex, hard-to-define human preferences into AI systems. While RLHF is well-established in language modeling, its application to image generation presents

unique opportunities. Unlike discrete language models that require policy gradient algorithms like PPO (Schulman et al., 2017), diffusion and flow-based models operate in a continuous space. This allows for more direct optimization methods. One such method is Reward-supported Flow Learning (ReFL) (Xu et al., 2023), which leverages the differentiability of flow models to directly backpropagate reward signals, proving highly efficient for visual alignment tasks.

To address the critical challenge of detail loss, we propose **UPER** (**U**nifying **P**ost-Training for **Per**sonalization). UPER is a post-training framework designed to enhance the detail consistency of any subject-driven generative model. Our framework consists of two core stages:

1. **Refined Supervised Fine-Tuning (SFT):** We introduce a data pre-processing pipeline that cleans reference images by removing confounding background information. While background removal itself is a known technique in object-centric generation (Chen et al., 2024; Song et al., 2024), we integrate it as a systematic SFT step to force the model to focus on subject-specific details.
2. **Reinforcement Learning (RL) with a Novel Reward Ensemble:** We design a composite reward function that balances text alignment, aesthetics, and a novel, patch-based reward metric specifically engineered to measure fine-grained subject consistency. This reward is optimized using the efficient ReFL algorithm, which we found to be more effective than DPO in our preliminary experiments.

We demonstrate UPER's effectiveness by applying it to the state-of-the-art OminiControl (Tan et al., 2024) model. Extensive automated, quantitative, and human evaluations confirm that UPER significantly improves detail preservation without compromising overall generation quality.

Our primary contributions are:

- A systematic, two-stage post-training framework (UPER) that significantly resolves the detail-consistency problem in personalized object generation by treating it as an alignment task.

- A new patch-based reward metric for subject consistency that leverages pre-trained vision encoders to capture fine-grained details, requiring no training on preference data.

- Extensive empirical validation, including a large-scale human study and comprehensive ablation experiments, showing that UPER achieves state-of-the-art subject fidelity and is overwhelmingly preferred by users over strong baselines.

## 2 RELATED WORK

### 2.1 REINFORCEMENT LEARNING FROM HUMAN FEEDBACK

Reinforcement Learning from Human Feedback (RLHF) has become a cornerstone for aligning AI systems, particularly Large Language Models (LLMs), with complex human values (Christiano et al., 2017). The standard process involves Supervised Fine-Tuning (SFT) on curated examples, followed by training a reward model (RM) on human preference data. Finally, a reinforcement learning algorithm optimizes the SFT model to maximize the score from the RM. While policy gradient methods like PPO (Schulman et al., 2017) are common, recent work has explored more sample-efficient alternatives like GRPO (Shao et al., 2024). Our work adapts this alignment paradigm to the continuous domain of image generation.

### 2.2 HUMAN FEEDBACK IN DIFFUSION MODELS

Integrating human feedback into diffusion models has become an active area of research, with several algorithmic families emerging to align models with preferences like aesthetic quality and semantic fidelity. One major branch of work adapts traditional reinforcement learning paradigms. This includes methods that use policy gradient algorithms like PPO (Black et al., 2023; Fan et al., 2023), which often introduce significant training complexity, and more direct fine-tuning approaches like Reward-supported Flow Learning (ReFL) (Xu et al., 2023), which leverage the model's differentiability to efficiently backpropagate a reward signal. A second branch seeks to simplify this process. Reward-Weighted Regression (RWR) (Lee et al., 2023) reframes alignment as a weighted supervised learning problem, while Direct Preference Optimization (DPO) and its variants (Rafailov

et al., 2023; Wallace et al., 2024) offer an elegant solution by bypassing the need for an explicit reward model altogether. While DPO is powerful, we found in preliminary experiments that the explicit, component-wise control offered by ReFL was more stable and effective for our specific multi-objective task. The ability to explicitly weight and balance different reward components (text, aesthetics, consistency) is crucial for navigating the complex trade-offs in our problem, a level of control that is less direct with DPO's implicit reward formulation. While these methods have proven effective for general T2I alignment, our work is the first to construct a reward ensemble specifically for the complex, multi-faceted task of detail-preserving personalized generation.

## 2.3 PERSONALIZED GENERATION

Personalized generation seeks to create images featuring a specific subject, style, or concept provided by a user. Early methods like Textual Inversion (Gal et al., 2022) and DreamBooth (Ruiz et al., 2023) achieved this through per-subject fine-tuning of a diffusion model on a few example images. While effective, these approaches are computationally intensive and require optimization for each new subject. More recent works, such as IP-Adapter (Ye et al., 2023) and our baseline OminiControl (Tan et al., 2024), have shifted towards using lightweight adapters for more efficient, zero-shot personalization. A parallel line of research, focused on high-fidelity object composition and editing, has also emerged. Works like AnyDoor (Chen et al., 2024), IMPRINT (Song et al., 2024), and Bifröst (Li et al., 2024) have explored sophisticated techniques for object manipulation, often involving segmentation. Our work draws inspiration from this latter line of research, specifically the principle of using background removal to isolate the subject. However, we position this not as a core novel contribution in itself, but as a crucial and systematic data refinement step within our broader alignment framework. The primary novelty of UPER lies in its two-stage post-training structure, which addresses the subsequent and more challenging problem of preserving the fine-grained details that even these advanced methods can struggle with.

## 3 METHOD

Our method, UPER, enhances personalized image generation through a two-stage post-training framework. The process begins with Supervised Fine-Tuning (SFT) to refine conditional focus, followed by Reinforcement Learning (RL) to optimize for a composite reward signal. The full pipeline is shown in Fig. 1 and detailed in Algorithm 1.

### 3.1 REWARD MODEL ENSEMBLE FOR PERSONALIZATION

Instead of training a monolithic reward model, we construct a composite reward by ensembling three specialized, pre-trained models. This approach allows us to precisely target the multi-faceted goals of personalized generation: text alignment, aesthetic quality, and, most critically, subject consistency. To measure semantic correspondence with the input prompt, we use the cosine similarity between CLIP ViT-L/14 embeddings of the generated image and the text, providing a standard, differentiable score for text alignment ($R_{\text{text}}$). For visual appeal, we employ the Human Preference Score v2 (HPS-v2) (Wu et al., 2023), a state-of-the-art aesthetic predictor trained on a large dataset of human preference choices, which yields a robust aesthetic score ($R_{\text{aes}}$). The cornerstone of our ensemble, however, is the reward for subject consistency ($R_{\text{sub}}$), which is designed to capture the fine-grained details that define a subject's identity.

#### 3.1.1 TEXT-PROMPT ALIGNMENT ($R_{\text{TEXT}}$)

We measure semantic correspondence between the generated image $I_{\text{gen}}$ and the prompt $P$ using CLIP ViT-L/14 embeddings (Radford et al., 2021): $R_{\text{text}} = \text{sim}(\text{CLIP}_{\text{img}}(I_{\text{gen}}), \text{CLIP}_{\text{text}}(P))$.

#### 3.1.2 AESTHETIC QUALITY ($R_{\text{AES}}$)

We use the Human Preference Score v2 (HPS-v2) (Wu et al., 2023), a state-of-the-art aesthetic predictor, to get a scalar score. HPS-v2 is trained on a large-scale dataset of human preference choices, making it robust against common failure modes and reward hacking. $R_{\text{aes}} = \text{HPS-v2}(I_{\text{gen}})$.

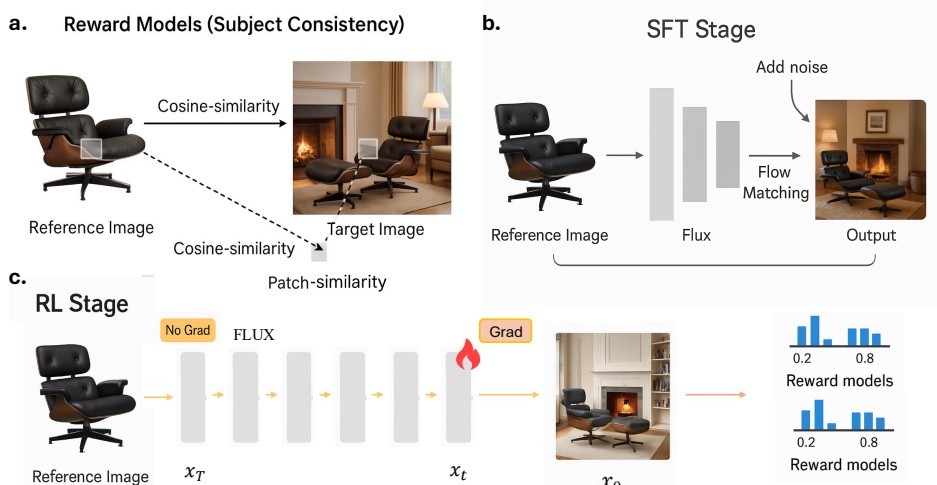

Figure 1: **Overview of the UPER Post-Training Pipeline.** The framework consists of two main stages: Supervised Fine-Tuning (SFT) and Reinforcement Learning (RL). **(a) Reward Model Ensemble:** We design a composite reward signal from three distinct, pre-trained components targeting text alignment ($R_{\text{text}}$), aesthetics ($R_{\text{aes}}$), and subject consistency ($R_{\text{sub}}$). The key innovation is our patch-based consistency metric, which uses a DINOv2 (Oquab et al., 2023) encoder to compute similarity at a local level. **(b) SFT Stage:** The base model is fine-tuned on a refined dataset where reference images have their backgrounds removed, forcing the model to learn a more precise subject-focused representation. **(c) RL Stage:** We use Reward-supported Flow Learning (ReFL) to align the model with the composite reward. The end-to-end differentiability of the reward models and the single-step flow prediction allows gradients to be backpropagated directly into the model's LoRA weights for efficient optimization.

### 3.1.3 SUBJECT CONSISTENCY ($R_{\text{SUB}}$)

To capture fine-grained details, we propose a patch-based reward, $R_{\text{sub}}$. The key insight is to use an encoder trained specifically for instance-level matching, rather than global semantic similarity. We choose DINOv2 (Oquab et al., 2023) for its strong performance on such tasks, as its self-supervised training objective encourages learning features that are robust to viewpoint changes while preserving identity. The computation is a three-step process. First, both the generated image $I_{\text{gen}}$ and the reference image $I_{\text{ref}}$ are decomposed into a grid of $N$ overlapping $224 \times 224$ patches with a stride of 112. Second, for each spatially corresponding patch pair $(p_k^{\text{gen}}, p_k^{\text{ref}})$, we extract their feature embeddings using the pre-trained DINOv2 encoder ($f_{\text{DINOv2}}$) and compute their cosine similarity:

$$\phi_k = \text{sim}(f_{\text{DINOv2}}(p_k^{\text{gen}}), f_{\text{DINOv2}}(p_k^{\text{ref}})). \tag{1}$$

Finally, the individual patch similarities are aggregated by taking their mean to produce the final subject consistency reward, $R_{\text{sub}} = \frac{1}{N}\sum_{k=1}^{N}\phi_k$. This patch-based approach is highly sensitive to local texture and pattern loss, which global metrics like CLIP similarity often miss.

### 3.2 POST-TRAINING PIPELINE

### 3.2.1 STAGE 1: SUPERVISED FINE-TUNING WITH REFINED CONDITIONING

The pre-training of our baseline model, OminiControl, utilizes the Subject-200K dataset. A critical observation is that the reference images in this dataset contain rich and often complex background information. As illustrated in Fig. 2(a), this creates a "conditioning noise" problem. For instance, when the model is tasked to learn the identity of the Eames lounge chair, it is simultaneously exposed to vastly different backgrounds—a cozy library in one image and a modern city view in another. This irrelevant background information can confound the model, forcing it to entangle subject features with background context and hindering its ability to learn a pure, disentangled representation of the subject's core attributes.

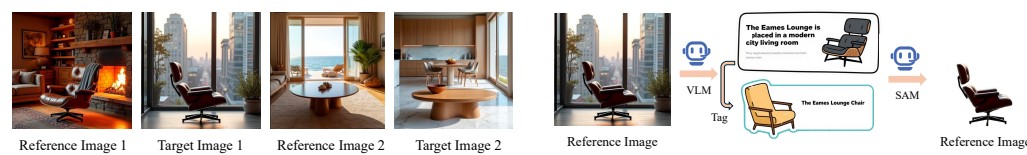

| | | | | | |
|---|---|---|---|---|---|
| Reference Image 1 | Target Image 1 | Reference Image 2 | Target Image 2 | Reference Image | Reference Image |

(a) Original data with noisy backgrounds.        (b) Our data refinement pipeline.

Figure 2: **SFT Data Refinement Pipeline.** (a) Original image pairs from Subject-200K exhibit "conditioning noise," where complex backgrounds interfere with subject learning. (b) Our pipeline first removes the background from the reference image and then uses a VLM to filter for high-quality pairs, ensuring the model focuses on core subject attributes.

---

**Algorithm 1** UPER Post-Training Framework

---

1: **Input:** Pre-trained model $\theta_0$, SFT dataset $D_{\text{SFT}}$, RL prompts $D_{\text{RL}}$.
2: **Hyperparameters:** SFT steps $T_{\text{SFT}}$, RL steps $T_{\text{RL}}$, learning rate $\eta$, LoRA rank $r = 4$.
3: Initialize LoRA weights for model $\theta_0$.
4: {— Stage 1: Supervised Fine-Tuning —}
5: **for** $t = 1$ **to** $T_{\text{SFT}}$ **do**
6:     Sample $(I_{\text{ref}}, I_{\text{target}}, P) \sim D_{\text{SFT}}$.
7:     Compute SFT loss $\mathcal{L}_{\text{SFT}}$ (e.g., flow matching loss).
8:     Update LoRA weights: $\theta \leftarrow \theta - \eta\nabla_\theta\mathcal{L}_{\text{SFT}}$.
9: **end for**
10: Let $\theta_{\text{SFT}} \leftarrow \theta$.
11: {— Stage 2: Reinforcement Learning —}
12: **for** $t = 1$ **to** $T_{\text{RL}}$ **do**
13:     Sample $(I_{\text{ref}}, P) \sim D_{\text{RL}}$.
14:     Generate image $I_{\text{gen}} \sim \pi_{\theta_{\text{SFT}}}(\cdot|I_{\text{ref}}, P)$.
15:     Compute rewards $R_{\text{text}}, R_{\text{aes}}, R_{\text{sub}}$.
16:     For each reward $R_i$, compute mean $\mu_i$ and std $\sigma_i$ over the batch.
17:     Normalize rewards: $\hat{R}_i \leftarrow (R_i - \mu_i)/(\sigma_i + \epsilon)$.
18:     Compute composite reward $R_{\text{composite}} = \sum w_i\hat{R}_i$.
19:     Compute RL loss $\mathcal{L}_{\text{RL}} = -R_{\text{composite}}$.
20:     Update LoRA weights: $\theta_{\text{SFT}} \leftarrow \theta_{\text{SFT}} - \eta\nabla_{\theta_{\text{SFT}}}\mathcal{L}_{\text{RL}}$.
21: **end for**
22: **Return:** Aligned model $\theta_{\text{RL}} = \theta_{\text{SFT}}$.

---

To address this information redundancy and improve the model's focus, we introduce a systematic data pre-processing and filtering pipeline for the SFT stage, as visualized in Fig. 2(b). This process is twofold. First, we apply a robust background removal model ('RMBG-1.4') to every reference image, segmenting the primary subject and placing it on a neutral white background. This step forces the model to learn the subject's identity from its intrinsic properties alone, free from confounding background signals. Second, to further enhance the quality and consistency of the training pairs, we employ a powerful Vision-Language Model, Qwen-VL (Wang et al., 2024), as a filter. For each pair, the VLM first identifies key visual attributes from the now-cleaned reference image (e.g., "Eames Lounge Chair," "black leather," "wood shell"). It then verifies whether these essential attributes are accurately present in the corresponding target image. Any pair that fails this cross-modal consistency check is discarded from the training set. This meticulous refinement process yields a high-quality SFT dataset that enables the model to develop a more robust and detailed conditional generation capability before the RL alignment stage.

### 3.2.2 STAGE 2: REINFORCEMENT LEARNING WITH DIFFERENTIABLE REWARDS

Following SFT, we use RL to align the model with our composite reward. We employ Reward-supported Flow Learning (ReFL) (Xu et al., 2023), where the reward signal is backpropagated directly through the single-step image prediction process. The RL loss is the negative of the composite reward: $\mathcal{L}_{\text{RL}} = -R_{\text{composite}}$. This end-to-end differentiable pipeline enables highly efficient alignment. The full process is detailed in Algorithm 1.

### 3.3 MITIGATING REWARD HACKING

An unconstrained $R_{\text{sub}}$ could encourage the model to simply copy-paste textures. We employ two strategies to mitigate this:

1. **Balanced Composite Reward:** We combine the reward components using weights determined via empirical sweeps: $R_{\text{composite}} = 0.2 \cdot \hat{R}_{\text{text}} + 0.2 \cdot \hat{R}_{\text{aes}} + 0.4 \cdot \hat{R}_{\text{sub}}$, where $\hat{R}$ denotes z-score normalization over the batch. This multi-objective landscape discourages over-optimization.

2. **Gradient Clipping:** To prevent the subject consistency term from dominating, we clip the gradient of the reward with respect to the generated image, $\nabla_{I_{\text{gen}}} R_{\text{sub}}$, with a threshold of $\tau = 0.2$.

## 4 EXPERIMENTS

### 4.1 EXPERIMENTAL SETUP

**Base Model.** We build UPER upon OminiControl (Tan et al., 2024), which is based on the FLUX.1-dev flow transformer model (Esser et al., 2024).

**Training Details.** We use LoRA (Hu et al., 2022) with rank 4. Training is done on 8 NVIDIA H100 (80GB) GPUs with an effective batch size of 32. We use the AdamW optimizer (Kingma & Ba, 2014) with a learning rate of 1e-4. The SFT stage runs for 5k iterations, and the RL stage for 2k.

**Datasets.** We use our refined version of Subject-200K (Tan et al., 2024) for SFT and the Dream-Booth dataset (Ruiz et al., 2023) for evaluation.

### 4.2 EVALUATION METHODOLOGY

**Baselines.** We compare UPER against OminiControl (our direct baseline) and IP-Adapter+FLUX (Ye et al., 2023), which represents a strong, widely-used method for subject-driven generation. This allows us to evaluate the specific gains from our post-training framework. **Quantitative Metrics.** We use Fréchet Inception Distance (FID) (Heusel et al., 2017) for overall image fidelity and CLIP Score (Radford et al., 2021) for text-prompt alignment. To specifically address the core challenge of this paper, we introduce **DINOv2-Sim**, which is the cosine similarity between the DINOv2 embeddings of the generated subject and the reference subject (both segmented from the background). This metric is designed to be a direct quantitative measure of subject consistency. **Automated & Human Evaluation.** For scalable assessment, we use GPT-4o to evaluate 750 generated image pairs on subject consistency, text alignment, and image fidelity. The cornerstone of our evaluation, however, is a large-scale human study. We collected over 1,000 responses from 105 unique participants in a blind, randomized pairwise comparison. The interface for this study, designed to elicit clear preferences on both quality and consistency, is shown in Fig. 3.

### 4.3 RESULTS AND ANALYSIS

**Quantitative and Automated Analysis.** Table 1 shows UPER consistently improves over baselines. The RL stage brings the most significant gain in DINOv2-Sim (+0.07 over SFT), confirming its effectiveness in enhancing subject consistency. The automated evaluation in Fig. 4(a) corroborates this, showing a major improvement in Subject Consistency as judged by GPT-4o, while maintaining strong Text Alignment and Image Fidelity.

**User Study Analysis.** The human evaluation (Fig. 4(b)) provides the most compelling evidence. When asked for their overall preference, users chose our final UPER-RL model over the baseline an overwhelming **89.3%** of the time. This near 9-to-1 preference ratio validates that by optimizing for detail consistency, we have addressed a primary pain point for users.

### 4.4 QUALITATIVE ANALYSIS

Beyond quantitative metrics, a qualitative examination of the generated images provides clear and intuitive evidence of UPER's effectiveness. In Fig. 5, we present a side-by-side comparison of our final UPER-RL model against the baseline for several challenging subjects. For the backpack,

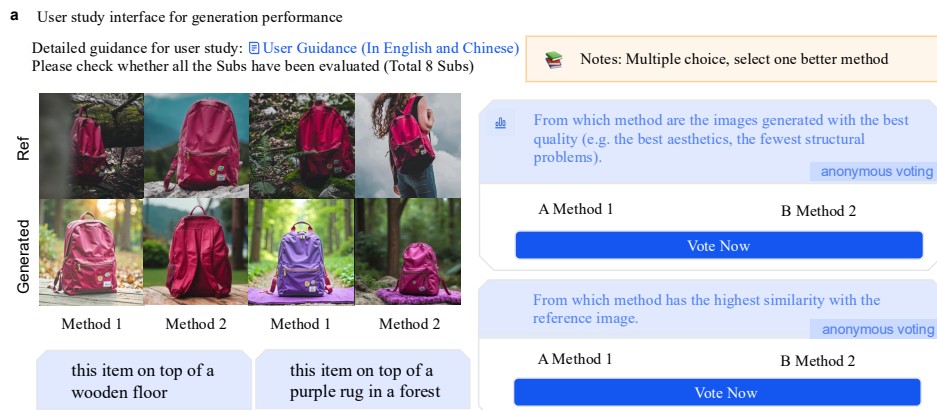

a   User study interface for generation performance

Detailed guidance for user study: 📖 User Guidance (In English and Chinese)
Please check whether all the Subs have been evaluated (Total 8 Subs)

Notes: Multiple choice, select one better method

Ref

Generated

Method 1   Method 2   Method 1   Method 2

this item on top of a wooden floor

this item on top of a purple rug in a forest

From which method are the images generated with the best quality (e.g. the best aesthetics, the fewest structural problems).

anonymous voting

A Method 1          B Method 2

Vote Now

From which method has the highest similarity with the reference image.

anonymous voting

A Method 1          B Method 2

Vote Now

Figure 3: **The interface for our human preference study.** Participants were presented with a reference image and a text prompt, along with two generated images from different models in a randomized order. They were asked to select the better image based on overall quality and subject consistency.

Table 1: Quantitative comparison. UPER demonstrates superior performance across all metrics, with significant gains in subject consistency (DINOv2-Sim) and image fidelity (FID).

| Method | FID $\downarrow$ | CLIP Score $\uparrow$ | DINOv2-Sim $\uparrow$ |
|---|---|---|---|
| IP-Adapter + FLUX | 239.12 | 0.782 | 0.65 |
| OminiControl (Baseline) | 156.12 | 0.824 | 0.71 |
| UPER-SFT | 134.12 | 0.830 | 0.78 |
| **UPER-RL (Ours)** | **130.12** | **0.831** | **0.85** |

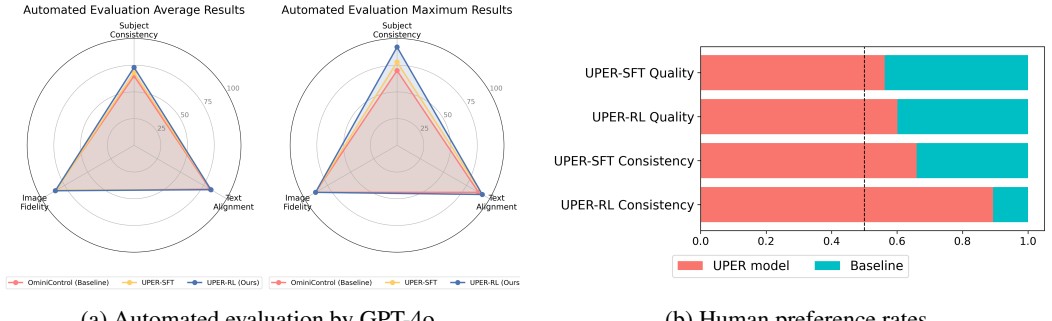

(a) Automated evaluation by GPT-4o.

(b) Human preference rates.

Figure 4: **Evaluation Results.** (a) The radar chart, normalized from 0 (worst) to 1 (best), shows UPER-RL's superior subject consistency. (b) The bar chart shows overwhelming human preference for UPER-RL over the baseline. Error bars denote 95% confidence intervals.

prompted with "a photo of this backpack in a forest," the baseline model generates a backpack of a different color, failing to preserve the original's distinct purple hue. Our model, however, maintains the correct color and texture. For the bowl, prompted with "a photo of this bowl in the snow," the baseline completely ignores the "Bon Appétit" text, a key identifying feature. UPER successfully reproduces this text, demonstrating superior alignment with human-salient details. Similarly, for the vase ("a photo of this vase on a wooden table"), UPER preserves the unique color gradient and glossy finish, while the baseline produces a duller, less accurate version. Finally, for the boots ("a photo of these boots on a cobblestone street"), UPER accurately reconstructs the intricate fringe details, which are heavily simplified by the baseline. These examples collectively illustrate that UPER consistently captures and renders the fine-grained, identity-defining characteristics that are crucial for high-fidelity personalization.

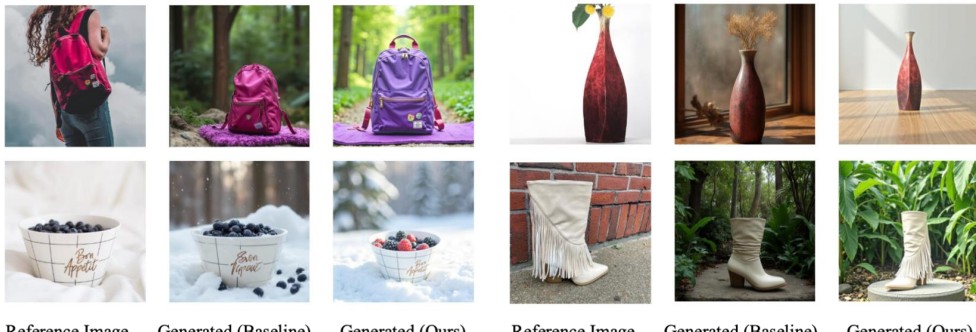

| Reference Image | Generated (Baseline) | Generated (Ours) | Reference Image | Generated (Baseline) | Generated (Ours) |

Figure 5: **Additional Qualitative Examples.** More comparisons showing UPER's superior detail preservation. Prompts from left to right, top to bottom: "a photo of this backpack in a forest", "a photo of this bowl in the snow", "a photo of this vase on a wooden table", "a photo of these boots on a cobblestone street".

## 4.5 ABLATION STUDIES

To dissect our framework's contributions and validate our design choices, we conducted a series of comprehensive ablation studies.

**Impact of SFT Data Refinement.** We first investigated the impact of our proposed SFT data refinement pipeline. As shown in Table 2, training a model without this pipeline (i.e., using the original Subject-200K dataset with noisy backgrounds) yields only a marginal improvement in subject consistency over the baseline. In contrast, our full SFT process, which uses cleaned reference images, leads to a substantial boost in both FID and DINOv2-Sim. This result empirically confirms our hypothesis that reducing conditioning noise by removing irrelevant backgrounds is a crucial first step for enhancing detail preservation.

Table 2: Ablation on SFT data refinement.

| SFT Variant | FID ↓ | DINOv2-Sim ↑ |
|---|---|---|
| OminiControl (Baseline) | 156.12 | 0.71 |
| UPER-SFT (w/o Refinement) | 145.53 | 0.73 |
| **UPER-SFT (Full)** | **134.12** | **0.78** |

**Contribution of Reward Components.** To understand the role of each component in our reward ensemble, we trained RL variants using only subsets of the rewards (Table 3). A model trained with only text alignment and aesthetic rewards ($R_{\text{text}} + R_{\text{aes}}$) failed to improve subject consistency, with its DINOv2-Sim score remaining at the SFT level. Conversely, a model trained with only the subject consistency reward ($R_{\text{sub}}$) achieved the highest consistency score but suffered from severe reward hacking, manifesting as unnatural texture repetition and a noticeable drop in text alignment (CLIP Score). This demonstrates that the full ensemble is necessary to achieve a synergistic effect, simultaneously improving consistency while maintaining quality and editability.

Table 3: Ablation on RL reward components.

| RL Reward | CLIP Score ↑ | DINOv2-Sim ↑ |
|---|---|---|
| UPER-SFT (No RL) | 0.830 | 0.78 |
| $R_{\text{text}} + R_{\text{aes}}$ only | 0.832 | 0.77 |
| $R_{\text{sub}}$ only | 0.815 | **0.86** |
| **Full Ensemble (Ours)** | **0.831** | 0.85 |

**Analysis of Reward Weights and Patch Encoder.** We tested alternative weightings for $R_{\text{composite}}$ and found our chosen weights (0.2, 0.2, 0.4) provided the best balance between consistency and

quality. We also compared DINOv2 with CLIP as the patch encoder for $R_{\text{sub}}$ (Table 4). DINOv2, which is self-supervised for fine-grained instance-level matching, significantly outperformed CLIP, which is trained for global semantic alignment. This highlights the importance of choosing a reward encoder whose training objective aligns with the desired fine-grained comparison task.

Table 4: Ablation on patch encoder for $R_{\text{sub}}$.

| Patch Encoder | DINOv2-Sim ↑ |
|---|---|
| CLIP ViT-L/14 | 0.81 |
| **DINOv2 ViT-g/14** | **0.85** |

## 4.6 COMPARISON WITH ADDITIONAL BASELINES

To further contextualize UPER's performance, we compare it with DreamBooth (Ruiz et al., 2023), a classic fine-tuning method, and a DPO-based (Wallace et al., 2024) variant of our own framework (Table 5). DreamBooth achieves excellent subject consistency but at the cost of requiring per-subject fine-tuning and offering limited text-based editability. Our DPO variant, which optimizes on preference pairs derived from our reward scores, was less stable during training for this multi-objective task and yielded slightly lower performance than our ReFL-based approach. This supports our choice of ReFL for its efficiency and effectiveness in this specific problem setting.

Table 5: Comparison with additional baselines.

| Method | Editability (CLIP) ↑ | Consistency (DINOv2) ↑ |
|---|---|---|
| DreamBooth | 0.795 | **0.87** |
| UPER (DPO-based) | 0.828 | 0.83 |
| **UPER (ReFL-based)** | **0.831** | 0.85 |

## 4.7 COMPARISON WITH STATE-OF-THE-ART GENERALIST MODELS ON HUMAN-ALIGNED BENCHMARKS

Meanwhile, we extended our evaluation to include OmniGen2 (Wu et al., 2025), a powerful, state-of-the-art generalist model known for its subject-driven generation capabilities. Crucially, to better evaluate the degree to which model personalisation aligns with human preferences, we evaluated its performance on the standardized DreamBench++ (Peng et al., 2024) benchmark, which provides robust criteria for assessing personalized image generation, alongside the original DreamBench.

We applied our Reinforcement Learning stage (using the same composite reward) to the publicly available OmniGen2 model to test the generalizability and effectiveness of our post-training alignment approach. As shown in Tab. 6, the results, obtained via automated scoring with GPT-4o, are clear. Our RL fine-tuning significantly boosts the subject consistency of OmniGen2 on both benchmarks (from 72.2 to 77.4 on DreamBench, and 84.0 to 88.0 on DreamBench++), with only a negligible impact on image fidelity and text alignment. This demonstrates that our specialist alignment framework is not only effective but also necessary for enhancing detail preservation, even when combined with powerful, large-scale generalist models. These results validate the broad applicability and necessity of the UPER framework.

Table 6: Automated evaluation (GPT-4o) of our RL alignment on the Omnigen2 model using Dream-Bench and the reviewer-suggested DreamBench++ benchmark. Our method consistently improves subject consistency.

| Model | Benchmark | Subject Consist. ↑ | Image Fidelity ↑ | Text Align. ↑ |
|---|---|---|---|---|
| Omnigen2 | DreamBench | 72.2 | 85.0 | 94.4 |
| Omnigen2 w. RL | DreamBench | **77.4** | 84.8 | **95.2** |
| Omnigen2 | DreamBench++ | 84.0 | **90.2** | 94.2 |
| Omnigen2 w. RL | DreamBench++ | **88.0** | 89.2 | **94.8** |

**Future Work.** The limitations of our current work point to several exciting directions for future research. To address the complexity of the reward ensemble, one could explore "reward distillation," where the knowledge from the three separate reward models is distilled into a single, efficient network. This would reduce the computational overhead during RL training. Another promising direction is to automate the reward weighting process, perhaps through meta-learning or a bandit-based approach, to find the optimal balance for different types of subjects or prompts dynamically. Finally, extending the UPER framework to other personalized generation tasks, such as video or 3D synthesis, where detail consistency is equally, if not more, critical, represents a significant and impactful area for future exploration.

## 5 CONCLUSION

In this paper, we introduced UPER, a two-stage post-training framework that significantly improves detail consistency in personalized image generation. By framing the problem as one of alignment and leveraging a novel, patch-based reward metric within an efficient RL framework, UPER successfully bridges the gap between model objectives and human perception. Our extensive evaluations, including comprehensive ablations and a large-scale user study, demonstrate that UPER produces more faithful and compelling personalized images that are overwhelmingly preferred by users. This work establishes a robust methodology for aligning generative models with nuanced, detail-oriented human preferences, paving the way for their use in high-fidelity creative and commercial applications where precision and faithfulness are non-negotiable.

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

## A  THE USE OF LARGE LANGUAGE MODELS(LLMS)

Large Language Models (LLMs) were only used to correct grammar errors and polish the writing. They were not involved in research ideation, experiment design, analysis, or other substantive contributions.

## B  LIMITATIONS

While our work achieves significant progress, it is subject to several limitations. First, the ultimate performance of the UPER framework is inherently dependent on the capabilities of the chosen base model (OminiControl in this paper). Although our post-training approach markedly improves detail consistency, it cannot fundamentally resolve certain intrinsic weaknesses of the base model, such as a limited understanding of complex spatial relationships or physical interactions. Second, our two-stage training pipeline, particularly the RL stage involving multiple reward models, introduces additional computational overhead and implementation complexity. The selection of reward weights requires empirical sweeps and may not be optimal for all subject types. Furthermore, despite employing strategies like gradient clipping to mitigate reward hacking, the risk of over-optimizing for a specific reward metric remains, which could lead to distortions in some aspects of the generated images. Lastly, our current evaluation focuses primarily on single-subject personalization; the framework's effectiveness in handling complex scenes with multiple interacting subjects remains an area for future investigation.

## C  BROADER IMPACT

**Positive Impact**    The technology proposed in this research holds the potential for positive impact across several domains. For artists, designers, and small businesses, it offers a powerful and efficient tool for creating highly customized visual content, such as product prototype visualizations, advertising materials, and personalized artwork, thereby lowering the barrier to professional content creation. In e-commerce and fashion, this technology could power applications like virtual try-on, offering consumers a more realistic and engaging shopping experience.

**Potential Risks and Mitigation**    Like all powerful generative technologies, the outcomes of this research carry a risk of misuse. The most significant concern is the potential for creating deceptive synthetic content ("deepfakes") to spread misinformation or for malicious purposes. While our research aims to enhance the fidelity of personalization, this capability is inherently a double-edged sword. We advocate for the continued development and deployment of robust synthetic media detection techniques to counter such risks. Moreover, the presence of copyrighted material and societal biases in the training data is a critical issue. The model might inadvertently replicate copyrighted elements or amplify biases inherent in the data. We believe future work must address the provenance and compliance of datasets and develop algorithms to identify and mitigate bias in generated content to ensure the responsible development and application of this technology.

## A   ADDITIONAL QUALITATIVE RESULTS

To further demonstrate the effectiveness of UPER, we provide additional qualitative comparisons in Fig. 6. These examples span a diverse range of subjects, including animals, toys, and household items, consistently showing UPER's superior ability to preserve fine-grained details, textures, and unique features compared to the OmniControl baseline.

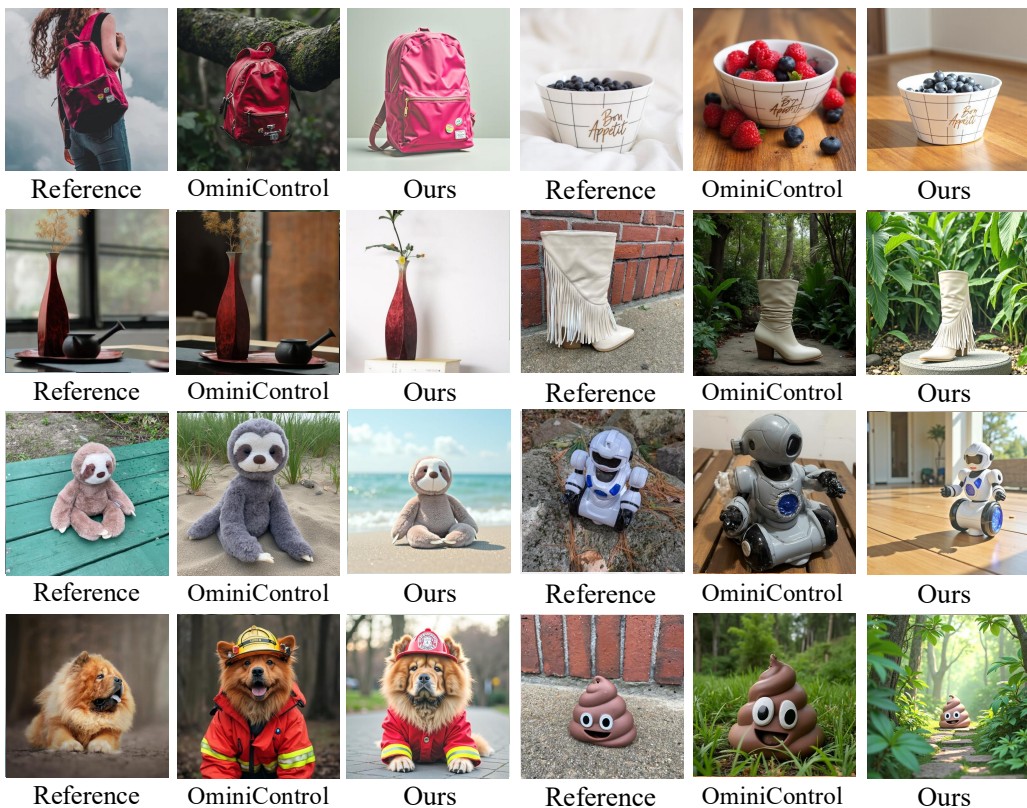

Figure 6: **Additional Qualitative Comparisons.** UPER consistently preserves subject-specific details such as the sloth's knitted texture, the robot's specific markings, the dog's fur pattern, and the unique shape of the poop emoji pillow, whereas the baseline often loses these fine-grained characteristics.

## B   FURTHER IMPLEMENTATION AND DESIGN DETAILS

### B.1   IMPLEMENTATION OF THE PATCH-BASED SUBJECT CONSISTENCY REWARD ($R_{sub}$)

As requested by reviewers, we provide a more detailed breakdown of our subject consistency reward calculation. The goal is to measure both global structural similarity and local detail fidelity, while preventing trivial copy-paste solutions.

1. **Global Similarity:** We first resize both the reference image $I_{\text{ref}}$ and the generated image $I_{\text{gen}}$ to the native resolution of our DINOv2 encoder (384x384 pixels). We extract their global CLS token embeddings and compute the cosine similarity. This provides a baseline score for overall structural correspondence.

2. **Patch-Based Similarity:** We then perform random spatial cropping on both images to extract $N$ corresponding patch pairs, each of size 384x384. For each pair, we compute the cosine similarity of their DINOv2 embeddings and average these scores across all $N$ pairs. This patch-level comparison is crucial for capturing fine-grained textures and patterns.

3. **High Similarity Penalty:** A key challenge in reward design is preventing "reward hacking." An unconstrained similarity metric could incentivize the model to simply output a slightly distorted version of the reference image. To mitigate this, we introduce a penalty for excessively high similarity scores. If the calculated similarity exceeds a certain threshold, the reward is penalized, encouraging the model to integrate the subject into a new context rather than just copying it.

The final $R_{sub}$ is a weighted combination of these components, creating a robust metric that aligns well with human perception of detail consistency.

## B.2 RATIONALE FOR REWARD WEIGHTS

The reward weights (0.2 for $R_{\text{text}}$, 0.2 for $R_{\text{aes}}$, and 0.4 for $R_{\text{sub}}$) were determined through empirical sweeps. Our initial approach used a more balanced distribution (e.g., 0.3, 0.3, 0.4). While this improved subject consistency, we observed instances of reward hacking from the aesthetic and text-alignment models, leading to minor artifacts. By down-weighting $R_{\text{text}}$ and $R_{\text{aes}}$, we found a better equilibrium that strongly preserved subject details without compromising overall image quality. We argue that the ratio between weights is more critical than their sum, as the overall magnitude can be absorbed by the learning rate. Our chosen weights represent the best-found trade-off for our task.

## B.3 QUALITATIVE ABLATION: DINOV2 VS. CLIP FOR $R_{sub}$ ENCODER

To illustrate why DINOv2 is superior to CLIP as a patch encoder for $R_{sub}$, we present a qualitative comparison in Fig. 7. While the CLIP-based reward model generates an image that is semantically correct at a high level (e.g., it produces "a robot" or "a bowl"), it fails to capture the specific, identity-defining details. The DINO-based reward, in contrast, successfully preserves fine-grained features such as the robot's blue chest light and wheel structure and the "Bon Appétit" text on the bowl. This is because DINOv2 is trained via self-supervision for instance-level matching, making its feature space inherently better suited for measuring the similarity of fine details.

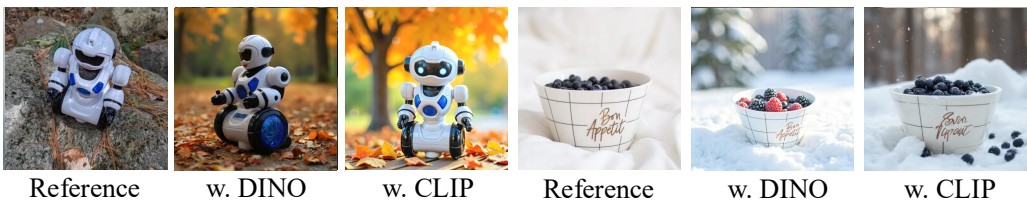

| Reference | w. DINO | w. CLIP | Reference | w. DINO | w. CLIP |

Figure 7: **DINOv2 vs. CLIP as Patch Encoder for $R_{sub}$.** The model trained with a DINOv2-based reward preserves fine-grained details (e.g., the robot's blue markings, the bowl's text). The model trained with a CLIP-based reward captures the high-level concept but loses these specific details, demonstrating the importance of using an encoder suited for instance-level matching.

## B.4 EXAMPLES OF VLM-FILTERED DATA PAIRS

In our SFT data refinement pipeline, the Qwen-VL model acts as a crucial filter to remove imperfectly paired data. Fig. 8 provides a concrete example of this process. The VLM is tasked with comparing two bookshelf images that share a similar overall structure. However, it correctly identified subtle but critical differences in their construction, the items on the shelves, and the wood grain pattern. By analyzing these fine-grained details, the model concludes that they are not the same object and assigns a similarity score of 0. This capability to discern between objects with high semantic similarity but low instance-level identity is vital for curating a high-quality SFT dataset, free from inconsistent or noisy pairs. For instance, another pair might be discarded if the reference image clearly shows a "red leather chair," but the target image, due to lighting or other factors, renders the chair as brown or orange.

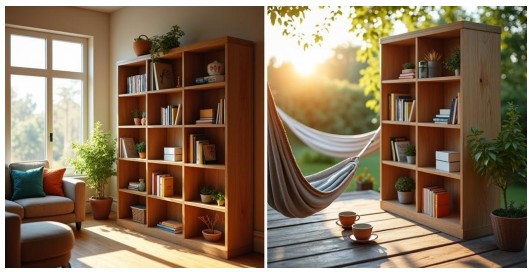

User: Are the bookshelves in the two pictures below identical?

VLM: While they share the same basic design (a light-colored wood, 2-column by 5-row cubby structure), there are numerous differences in their construction, the items they hold, and even the wood grain itself. **They are definitively two different objects.**

User: Are the bookshelves in the two pictures below identical? *Rate this pair. 0 indicates they are definitely not the same object, while 5 indicates they are certainly the same object.*

VLM: While they share the same basic design (a light-colored wood, 2-column by 5-row cubby structure), there are numerous differences in their construction, the items they hold, and even the wood grain itself. **They are definitively two different objects. Therefore, the pair will be rates as 0.**

Figure 8: **Example of VLM-Based Data Filtering.** The VLM is tasked with determining if the two bookshelves are identical. Despite sharing a similar high-level design (a 2-column, 5-row wooden structure), the VLM astutely identifies fine-grained inconsistencies in its construction, the specific items it holds, and even the wood grain. Consequently, it rates the pair as 0, signifying they are different objects. This showcases the VLM's crucial role in filtering out subtly mismatched pairs to ensure the quality of our SFT dataset.

## C    QUALITATIVE ANALYSIS OF UPER ON OMNIGEN2

In response to the reviewer's feedback, we not only added quantitative comparisons with the state-of-the-art OmniGen2 model but also provided qualitative examples in Fig. 9. These visualizations demonstrate the effectiveness of our RL alignment stage when applied to a strong, existing generalist model. The results on both the standard DreamBench benchmark and the more challenging DreamBench++ benchmark show that our method consistently enhances subject consistency and detail preservation.

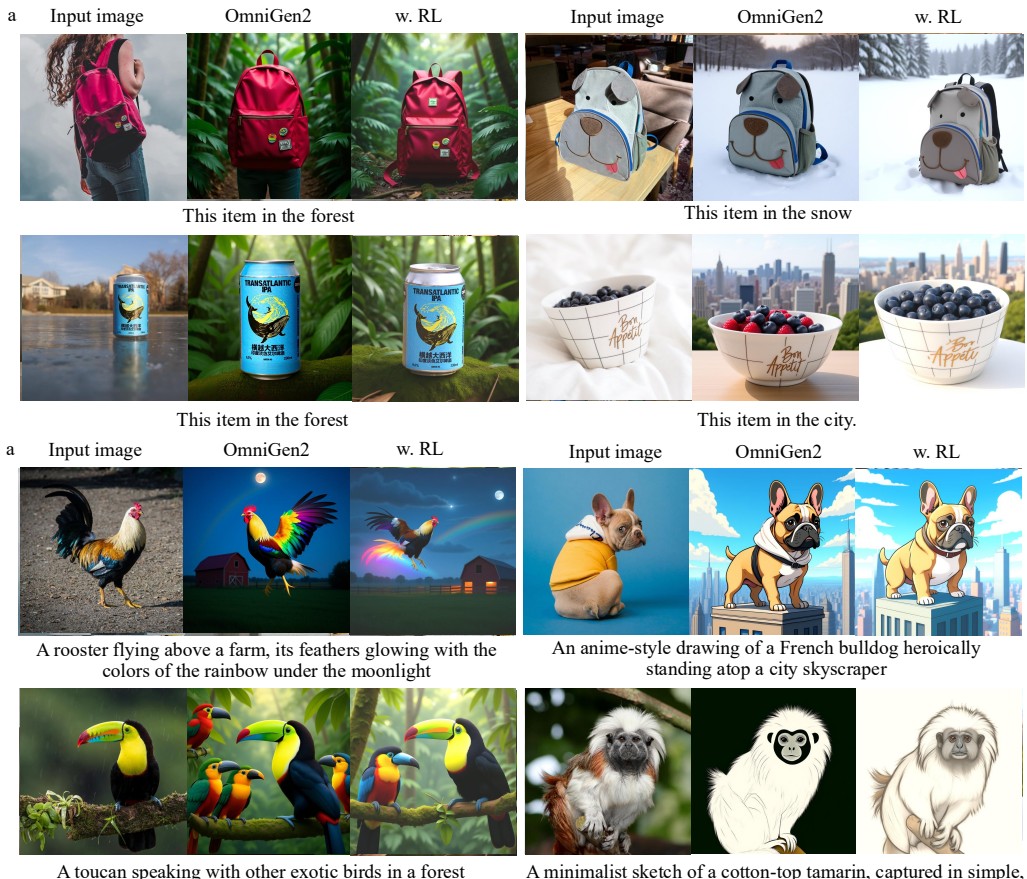

Figure 9: **Qualitative comparison of our RL alignment applied to Omnigen2.** The figure is divided into two parts based on the evaluation benchmark. **(a)** Results on the DreamBench benchmark. Our RL alignment (w. RL) significantly improves detail preservation. For instance, it correctly renders the pins on the pink backpack, preserves the "Transatlantic IPA" text on the can, and maintains the "Bon Appétit" script on the bowl, details which the base OmniGen2 model struggles with. **(b)** Results on the DreamBench++ benchmark, which features more complex and stylistic prompts. Our RL alignment successfully enhances subject identity. For example, it generates a more faithful anime-style French bulldog while preserving the subject's core features, and creates a more vibrant and detailed rooster that better matches the fantastical prompt. These examples show that the UPER framework serves as a general-purpose post-training solution to improve subject fidelity for state-of-the-art models.