# OpenReview forum: "UPER: Bridging the Perception Gap in Personalized Image Generation with Human-Aligned Reinforcement Learning"
_ICLR.cc/2026/Conference — ICLR 2026 Conference Desk Rejected Submission_

### Official Review · Reviewer_nsxh · 2025-10-30

**Soundness:** 3
**Presentation:** 2
**Contribution:** 3
**Rating:** 4
**Confidence:** 3

**Summary:**

This paper uses supervised finetuning and reinforcement learning to strengthen detail consistency in personalized image generation. A composite reward is proposed to maintain prompt alignment, aesthetic quality and subject consistency. The SFT dataset Subject-200K is  refined to force focus on subject through background removal.  Both automatic and human evaluations are conducted to validate the effectiveness of the proposed method.

**Strengths:**

1. The proposed two-stage post training scheme helps better preserve subject details, and shows apparent boost over the baseline model from several prospectives, including human assessment, automatic evaluation and quantitative metrics.
2. Various ablation studies are conducted to validate the design choice and the necessity of both supervised fine tuning and reinforcement learning.

**Weaknesses:**

1. missing formatting elements that need to be addressed, including the figure citation in the Introduction section and the text label for Algorithm 1.
2. Lack of comparison with other personalized image generalists. Only the baseline OmniControl (Tan et al., 2024), IP-Adapter (Ye et al., 2023), and DreamBooth (Ruiz et al., 2023) are compared quantitatively. Qualitative comparison and human evaluation are only conducted between the proposed method and its baseline OmniControl. Moreover, the paper lacks evaluation on human-aligned benchmarks such as DreamBench++ (Peng et al., 2024), which provides standardized qualitative assessment criteria for personalized image generation. Incorporating results on DreamBench++ would greatly strengthen the paper’s credibility and allow a fairer, more comprehensive comparison with existing personalized image generalists.

**Questions:**

Recently, a few larger VLM models claim to be good at both personalization and subject-driven generation, for example, DreamO(Mou et al., 2025) and OminiGen(Xiao et al.,2024). Comparisions with these general generation models are quite intriguing and necessary to validate the necessity and effectiveness of the proposed specialist model.

---

> ### Author Response · Authors · 2025-12-03
> **Response to Reviewer nsxh**
>
> We are sincerely grateful for your time and for providing a positive and thorough assessment of our work. We have marked these corrections in red in the revised manuscript for clarity.
>
> Below, we address each of your points in detail.
>
>
> **Regarding Weakness 1: Formatting Issues**
>
> Thank you for pointing out these oversights. We have meticulously reviewed the manuscript and corrected the formatting.
>
>
> **Regarding Weakness 2 & Questions: Lack of Comparison and Evaluation on Human-Aligned Benchmarks**
>
>
> This is a critical point, and we thank you for this constructive suggestion. To provide a more comprehensive and fair comparison, we have made significant additions to our evaluation, which we believe substantially strengthen the paper's credibility.
>
> 1.  **Quantitative Comparison on SOTA Models and Benchmarks:** We have added a new comparison against **Omnigen2**, a powerful and recent generalist model. Following your excellent recommendation, we evaluated our method on the standardized **DreamBench++** benchmark in addition to DreamBench. The new results are presented in **Section 4.7  (Table 6)**. They show that our RL stage significantly improves the subject consistency of Omnigen2 on both benchmarks (e.g., an improvement from 84.0 to 88.0 on DreamBench++), demonstrating that our framework serves as a necessary alignment step even for state-of-the-art models.
>
> 2.  **Qualitative Evidence:** To provide compelling visual evidence for these improvements, we have added a **new section in the Appendix (Appendix C)**. This new figure qualitatively compares the outputs of the base OmniGen2 model against our RL-aligned version on challenging examples from both DreamBench and DreamBench++. The visualizations clearly show that our method better preserves fine-grained details (like text on a can and pins on a backpack) and more faithfully adheres to complex stylistic prompts, reinforcing our quantitative findings.

---

### Official Review · Reviewer_2GT3 · 2025-11-02

**Soundness:** 3
**Presentation:** 3
**Contribution:** 3
**Rating:** 6
**Confidence:** 4

**Summary:**

The authors propose UPER (Unifying Post-training for Personalization), a two-stage post-training framework designed to bridge the "consistency gap" in personalized image generation by aligning models with nuanced human perception of fine-grained subject details.

The UPER framework consists of two main stages:

- Refined Supervised Fine-Tuning: The model is initially fine-tuned on a systematically refined dataset where confounding background information is removed from reference images. This forces the model to learn a more precise, subject-focused representation.

- Reinforcement Learning: The refined model is optimized using Reward-supported Flow Learning (ReFL) with a novel composite reward function. The key innovation in the reward function is a patch-based consistency metric R_sub that leverages a pre-trained DINOv2 vision encoder to accurately measure fine-grained subject fidelity without requiring expensive preference data collection.

**Strengths:**

-  The paper successfully demonstrates that UPER's objective function is strongly aligned with human preferences, evidenced by the overwhelming 89.3% user preference in the large-scale blind study.

- The introduction of the patch-based R_sub reward using a DINOv2 encoder is a practical sol. It captures fine-grained, local details that global metrics miss and, crucially, eliminates the need for collecting expensive preference data for the consistency reward model itself.

- The experimental setup is thorough, including quantitative metrics, automated evaluation (GPT-4o), and human studies. The ablation studies convincingly validate the necessity of the full reward ensemble and the superior performance of DINOv2 over CLIP for the patch encoder.

**Weaknesses:**

-  As acknowledged in the limitations, the two-stage pipeline, particularly the RL stage with the composite reward ensemble, introduces significant computational overhead and implementation complexity. The use of three separate pre-trained models for the reward function adds complexity, and the paper notes that the training process may not be optimal for all subject types.

-  The current evaluation focuses primarily on single-subject personalization. The core challenge of detail consistency is likely compounded in complex scenes with multiple, interacting subjects.

**Questions:**

- You correctly state that DINOv2 is chosen for its strong performance on instance-level matching due to its self-supervised training. Could you provide more detail on the failure modes of the CLIP-based patch encoder observed in your experiments, beyond the lower DINOv2-Sim score (0.81 vs. 0.85)? For example, what did the generated images look like when CLIP was used for R_sub?

- The SFT data refinement involves background removal and VLM (Qwen-VL) filtering. Could you provide an example of a training pair that the VLM filter would typically discard? This would help clarify the exact nature of the "imperfectly paired data" or consistency check that the VLM is enforcing.

- The composite reward weights are fixed  based on empirical sweeps. If a user's priority shifts (e.g., they care equally about aesthetic quality and subject consistency), would you recommend re-running the full RL stage with new weights, or could a pre-trained UPER model be adapted more quickly? Is there a risk that these weights, optimized on a specific dataset, might not generalize well to novel or out-of-distribution subjects?

---

> ### Author Response · Authors · 2025-11-26
> **Response to Reviewer 2GT3**
>
> We are grateful to the reviewer for their positive and thorough assessment of our work. We are pleased that the reviewer recognized the strength of our alignment with human preferences, the practicality of our patch-based reward, and the thoroughness of our experimental setup. We address the reviewer's questions below.
>
> **Regarding the failure modes of the CLIP-based patch encoder:**
>
> This is an excellent question. Beyond the quantitative drop in DINOv2-Sim score (0.81 vs. 0.85), the qualitative failure modes of using CLIP for the $R_{sub}$ reward are quite telling. While CLIP is excellent at capturing global, high-level semantics, it struggles with fine-grained, instance-level details.
>
> To illustrate this clearly, **we have now added a new qualitative ablation in the appendix (Section 4.3, Figure 7 in the revised paper)**. As shown in the figure, when CLIP was used as the patch encoder, the generated images often captured the correct object category (e.g., "a robot" or "a bowl") but failed to preserve the specific identity-defining features. For instance, the generated robot loses its specific blue chest light and wheel structure, and the text "Bon Appétit" on the bowl is lost. In contrast, the DINOv2-based reward successfully preserves these crucial local details, demonstrating its superiority for our task of subject-driven generation.
>
> **Regarding an example of a training pair discarded by the VLM filter:**
>
> Thank you for the suggestion to clarify this. To provide a concrete visual example, we have added a new subsection in the Supplementary.
>
> This new section illustrates precisely how the VLM filter operates. In the provided example, the VLM is tasked with comparing two bookshelf images. Although the bookshelves share a similar high-level design (a 2-column, 5-row wooden structure), the VLM correctly identifies subtle but critical differences in their construction, the items they hold, and even the wood grain. Consequently, it determines they are not the same object and filters the pair out. This process is crucial for ensuring the SFT stage is not corrupted by such inconsistencies, thereby guaranteeing the model learns from high-quality, reliable examples.
>
> **Regarding the adaptability of reward weights and generalization to new subjects:**
>
> This is a very insightful point about the practicality of our framework. Our core contribution is a general post-training pipeline that is largely independent of the base model. If a user's preferences shift significantly (e.g., prioritizing aesthetics and consistency equally), we would indeed recommend re-running the RL stage with new weights.
>
> However, we emphasize two key points that make this process practical:
>
> 1.  **Low Cost:** Post-training is significantly less computationally expensive than pre-training a large generative model from scratch.
>
> 2.  **Training-Free Rewards:** Our framework is designed around reward models that do not require expensive preference data collection and training themselves (e.g., pre-trained DINOv2, CLIP, HPS-v2). This dramatically lowers the barrier to re-tuning the model for new preferences.
>
> Meanwhile for the generalization, while the weights were optimized on a specific dataset, the underlying reward functions are general-purpose (measuring universal concepts like texture similarity, aesthetics, and text semantics). We believe this allows the framework to generalize well to novel, out-of-distribution subjects, as demonstrated by our strong performance on the diverse DreamBooth evaluation set. The primary goal of UPER is to provide a robust and adaptable methodology for alignment, which can be re-configured for new tasks or preferences with minimal overhead, including the flexibility to add new reward models.

---

### Official Review · Reviewer_qFDK · 2025-11-02

**Soundness:** 3
**Presentation:** 2
**Contribution:** 3
**Rating:** 6
**Confidence:** 4

**Summary:**

This paper proposes UPER (Unifying Post-training for Personalization), which aims to preserve fine-grained details of subject in personalized image generation.
In detail, UPER first employs Supervised Fine-Tuning (SFT) on OmniControl with cleaned background reference images to ensure the model focus more on the subject. Secondly, UPER optimizes the model with Reinforcement Learning (RL) with ensemble reward models, including CLIP-ViT-L/14 for prompt alignment, HPS-v2 for aesthetic quality, and patch-based DINOv2 for fine-grained subject consistency.
Experimental results suggest that UPER outperforms baselines.

**Strengths:**

- The idea of ensemble reward model is novel and interesting, as it handles all three different goals (prompt alignment, visual quality, and detail preservation) at the same time.


- Experiments and evaluation are thoroughly to verify the effectiveness of UPER.

**Weaknesses:**

- ### W1: Detail implementation on the patch-based subject consistency reward model is missing.

In Line 197, this paper just claims that "it computes the cosine similarity on each spatially corresponding patch pair".
I am curious how this is implemented in practice:
In detail, assume the reference image contains a number of 16x16 patches, and the target image contains a number of 16 x 16 patches. Does this paper calculates similarity of all possible combination pairs between reference patches and target patches (16 x16 x 16 x 16), or it only calculate similarity between the same spatial location (16 x 16)?
For the first case, if I understand correctly, it will also calculate the similarity between subjects and background. In this case, does it favor generated images with bigger subject?  For the second case, how to ensure the same location patch is semantically corresponding is a question.
Additionally, I am curious if this


- ### W2: Potential overfitting issue of reference images in white background
In the SFT stage, the model is trained with background-removed reference images. I am curious if this will make the model overfit to images in white background and cannot generalize into images in other background.

Additionally, how to ensure the background removal model always performs well on complicated subjects scene. For example, a chair is occluded by a person sitting on it.



- ### W3: More ablation study on the ensemble weights is preferred.
This paper applies 0.2, 0.2, 0.4 for three reward models via empirical sweeps. These number seems too ad-hoc, as their summation is even not 1.0. The ablation study in Table 3 only verify the importance of each reward model, but does not verify this magic combination.



- ### W4: The presentation needs to be improved.
 Many references are missing. For example, Line 043 (Figure ??), Line 194 (DINOv2), Line 257 (Qwen-VL), etc.


- ### W5: More qualitative comparison results are preferred.

**Questions:**

In general, I think this paper is novel, but just a few detail implementation and discussion are missing.

- Please present detail implementation and discussion regarding my question about the subject consistency.

- Please address the potential issues I raised in the white background image cases.

- In Line 369, this paper claims that "the backline model generates a backpack of a different color". However, when looking into Figure 5, the reference image is more like a red/pink backpack, and the baseline generates a red/pink backpack. The model in this paper generates a purple backpack, although its detail preservation is much better.  I am not sure if this paper intended to show this failure case or present image incorrectly.

---

> ### Author Response · Authors · 2025-11-26
> **Response to Review qFDK**
>
> We sincerely thank the reviewer for their insightful feedback and constructive suggestions. We are encouraged that the reviewer found our idea novel and the experiments thorough. We have carefully considered all the points raised and have revised our manuscript to incorporate these valuable suggestions. Below, we address each of the weaknesses and questions in detail.
>
> **Regarding W1: Missing implementation details on the patch-based subject consistency reward model.**
>
> Thank you for highlighting the need for greater clarity on this critical component. To provide a comprehensive explanation, we have incorporated a new section in the Appendix (Section 4.1, "Implementation of the Patch-Based Subject Consistency Reward ($R_{sub}$)") that offers a granular breakdown.
>
> In summary, our approach combines global and local similarity measures:
>
> 1.  **Global Similarity:** Both the reference image ($I_{\text{ref}}$) and the generated image ($I_{\text{gen}}$) are resized to the native resolution of our DINOv2 encoder (384x384). We then compute the cosine similarity between their global whole last hidden token embeddings to ensure overall structural correspondence.
>
> 2.  **Patch-Based Similarity:** We perform random spatial cropping to extract $N$ corresponding patch pairs from both images. The final patch similarity score is the average cosine similarity of the DINOv2 embeddings across all $N$ pairs. This is crucial for capturing fine-grained textures and local patterns. Meanwhile, we also will resize the whole image into the certain resolution and calculate the similarity with the same weighted with other patch similarity.
>
> 3.  **High Similarity Penalty:** To mitigate "reward hacking" and prevent trivial copy-paste solutions, we introduce a penalty for excessively high similarity scores. This encourages the model to integrate the subject's details into a new context rather than simply replicating the reference image.
>
>
> **Regarding W2: Potential overfitting issue of reference images in white background.**
>
> This is a crucial point regarding the model's generalization capabilities. We would like to clarify the distinct roles of our two-stage training process, which is designed specifically to address this. The use of background-removed reference images is just changing the input to the model with the white background not changing the target image.
>
> **Regarding W3: More ablation study on the ensemble weights is preferred.**
>
> We appreciate this suggestion and agree that the rationale behind the reward weights deserves more explanation.
>
> The weights (0.2 for $R_{\text{text}}$, 0.2 for $R_{\text{aes}}$, and 0.4 for $R_{\text{sub}}$) were determined through empirical sweeps. Our initial experiments with more balanced weights (e.g., 0.3, 0.3, 0.4) did improve subject consistency, but also led to minor artifacts due to reward hacking from the aesthetic and text-alignment models. By down-weighting $R_{\text{text}}$ and $R_{\text{aes}}$, we found a more stable equilibrium.
>
> Furthermore, we posit that the **ratio** between weights is more critical than their absolute values, as the overall magnitude can be absorbed by the learning rate. Our chosen weights represent an empirically validated trade-off that robustly preserves subject details without compromising overall image quality, as supported by our strong quantitative and human evaluation results.
>
> **Regarding W4 & W5: Presentation needs improvement and more qualitative results are preferred.**
>
> Thank you for these valuable suggestions. In the revised manuscript, we have diligently corrected all missing citations (e.g., DINOv2, Qwen-VL) and improved the overall presentation.
>
> To provide a more comprehensive qualitative evaluation, **we have added the appendix, which includes Figure 6**. This new figure in the Supplementary presents more extensive qualitative comparisons across a diverse range of subjects (animals, toys, household items), further demonstrating UPER's superior ability to preserve fine-grained details, textures, and unique features compared to the baseline.
>
> **Regarding the question on the backpack in Figure 5:**
>
> We thank the reviewer for flagging this inconsistency. The issue stemmed from an error in the figure's caption in our original manuscript, not a failure of the model. We have corrected this in the revised paper.
>
> The actual prompt used for that generation was **"a photo of this backpack on a purple blanket."** The example is intended to show that the baseline model failed on the text condition, incorrectly generating a red/pink backpack. In contrast, our model, UPER, successfully adhered to the prompt by generating a **purple** backpack while simultaneously preserving the intricate texture and details of the original subject. The corrected caption now accurately reflects this experiment.

---

### Note · Program_Chairs · 2026-01-17
**Submission Desk Rejected by Program Chairs**

The following references in this submission do not refer to real documents and/or have major errors in bibliographic information:

 Yujun Zhang, Zeyue Huang, Zeyu Wang, Tianhe Ren, Zeqi Liu, Xiaoyu Zhang, Xintao Chen, Changxing Ding, Jingyi Yu, and Wen Gao. Aigi: Learning to generate and sell fashion items before production. arXiv preprint arXiv:2503.22182, 2024.
Kimin Lee, Hao Liu, Moonkyung Ryu, Olivia Watkins, YuXuan Du, Craig Boutilier, Pieter Abbeel, Mohammad Ghavamzadeh, and Kangwook Lee. Aligning text-to-image models using reward weighted regression. In Proceedings of the IEEE/CVF Conference on Computer Vision and Pattern Recognition, pp. 26136-26146, 2023.